# GraphGPT: Graph Learning with Generative Pre-trained Transformers

## Abstract

We introduce *GraphGPT*, a novel model for Graph learning by self-supervised Generative Pre-training Transformers. Our model transforms each graph or sampled subgraph into a sequence of tokens representing the node, edge and attributes reversibly using the Eulerian path first. Then we feed the tokens into a standard transformer decoder and pre-train it with the next-token-prediction (NTP) task. Lastly, we fine-tune the GraphGPT model with the supervised tasks. This intuitive, yet effective model achieves superior or close results to the state-of-the-art methods for the graph-, edge- and node-level tasks on the large scale molecular dataset PCQM4Mv2, the protein-protein association dataset ogbl-ppa and the ogbn-proteins dataset from the Open Graph Benchmark (OGB). Furthermore, the generative pre-training enables us to train GraphGPT up to 400M+ parameters with consistently increasing performance, which is beyond the capability of GNNs and previous graph transformers. The source code and pre-trained checkpoints will be released soon to pave the way for the graph foundation model research, and also to assist the scientific discovery in pharmaceutical, chemistry, material and bio-informatics domains, etc.

## 1 Introduction

After the breakthrough of deep learning in 2012 (Krizhevsky et al., 2012), the computer vision (CV) and natural language processing (NLP) communities prosper till now. Graph community also benefits from this paradigm shifting from traditional machine learning to deep learning with the proposal of various graph neural networks (GNNs) (Kipf & Welling, 2017; Hamilton et al., 2017; Zhang & Chen, 2018; Wu et al., 2020).

Nowadays, both CV (Dosovitskiy et al., 2021; Liu et al., 2021) and NLP (Devlin et al., 2019; Radford et al., 2018) adopt the transformer backbone as the canonical architecture, and expand the model scale to billions or even hundreds of billions of parameters (Liu et al., 2021; Brown et al., 2020). These large models are trained with web-scale data (Schuhmann et al., 2022; Touvron et al., 2023), and achieve performance superior to human beings in many benchmark datasets (Deng et al., 2009; Wang et al., 2019b;a). These works also boost the commercial applications like Midjourney (Midjourney, 2023) and ChatGPT (Open-AI, 2023). However, GNNs are still suffering from the over-smoothing (Rusch et al., 2023) and over-squashing (Alon & Yahav, 2021) problems, which limit them to be expanded to larger scale and trained with larger amount of graph data.

In recent years, there are many works that tried to utilize transformers to model the graph data, and obtained good performance in certain graph datasets (Ying et al., 2021; Kim et al., 2022; Luo et al., 2023; Müller et al., 2023). Nevertheless, they often use complex tricks to handcraft features to encode structure information explicitly, and then inject them into either the input or the attention layers. These tricks restrain the generalization beyond specific datasets. In addition, these graph transformers (GTs) often perform well in graph-level tasks, but not applicable for the edge-/node-level tasks (Müller et al., 2023). Last but not least, self-supervised generative pre-training is the key of the success of GPTs (Radford et al., 2018), but it is hardly incorporated in the GTs (Min et al., 2022; Müller et al., 2023).

In this work, we propose *GraphGPT*, a novel model for graph learning. It consists of three non-trivial components: 1) transforming the (sub)graphs into a reversible sequence of tokens via the Eulerian path, 2) pre-training a transformer decoder using the NTP task, and 3) fine-tuning the transformer

with any supervised graph tasks. GraphGPT is able to overcome the disadvantages of GNNs and Graph Transformers, and obtains state-of-the-art (SOTA) results in graph-/edge-/node-level tasks. Our contributions are summarized as follows:

- GraphGPT employs (semi-)Eulerian path[1] to transform the graphs into the sequences of tokens, which ensures the transformation is lossless, reversible and optimal. Together with subgraph sampling and node identity encoding techniques, we can transform graphs of various sizes into sequences of tokens.

- GraphGPT is generatively pre-trained with the NTP task, which brings 3 advantages: $a$) learning the graph's structural and semantic information without relying on handcrafted features or tailored transformer architectures; $b$) scaling up to 400M+ parameters with consistent performance improvement; $c$) possessing the capability to generate graphs.

- GraphGPT utilizes a novel approach to format graph-/edge-/node-level tasks, making them compatible with the transformer decoder architecture and enabling them to fully benefit from the generative pre-training.

- Through extensive experiments on the public OGB datasets, GraphGPT achieves SOTA results in graph-level and edge-level tasks, and the performance in node-level tasks approaches the SOTA level.

## 2 APPROACH

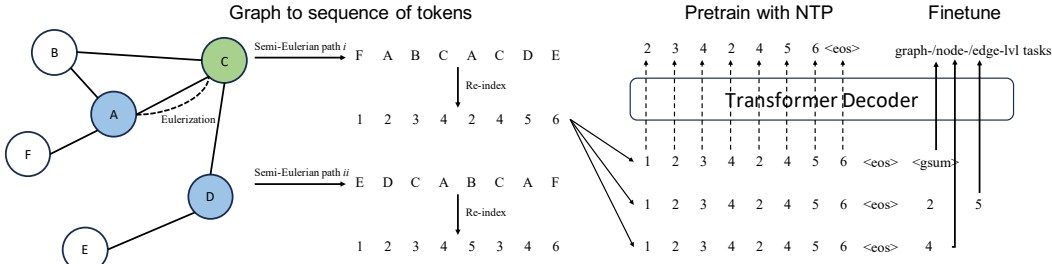

Figure 1: Model overview. The left part shows the procedure to convert a (sub)graph into a sequence of tokens. The dashed line indicates the duplicated edge added to the original graph to make it (semi-)Eulerian. The right part shows the pre-training with the self-supervised NTP task and fine-tuning on the supervised graph-/edge-/node-level tasks. We use edge $(A, D)$ as an example for the edge-level task, and node $C$ for the node-level task.

### 2.1 OVERVIEW

In our GraphGPT model, we transform the given graph into a sequence of tokens in a reversible manner, ensuring that the graph and the resulting sequence are equivalent. Then we utilize a standard transformer backbone, such as Llama from Touvron et al. (2023), to process the sequence data and train it using the self-supervised NTP task. Finally, we fine-tune the transformer with the supervised tasks such as graph classification/regression, link prediction and node classification.

### 2.2 GRAPH TO SEQUENCE OF TOKENS

In this section, we show how to convert a graph into a sequence of tokens. For small graphs, *e.g.*, molecular graphs, we convert them to sequences as in Sec. 2.2.1. Big graphs, *e.g.*, graphs with hundreds of thousands of nodes, millions or even billions of edges and many node or edge attributes, are processed differently. We first sample subgraphs (see Sec. 2.2.2) with the node identity information preserved (see Sec. 2.2.3), and then transform them into sequences in accordance with Sec. 2.2.1.

---

[1]For a quick recap, a graph with every node of even degree is Eulerian, and with exactly two nodes of odd degree is semi-Eulerian.

### 2.2.1 SERIALIZING GRAPHS WITH (SEMI-)EULERIAN PATH

We aim to find a lossless and reversible method to serialize (sub)graphs into sequences of tokens. This requires all the nodes and edges shall be represented in the sequence. Therefore, a path that transverses all the edges and nodes at least once would be a candidate.

This type of path is studied in the literature as the Chinese Postman Problem six decades ago, and it can be solved in polynomial time (Mei-Ko, 1962; Edmonds & Johnson, 1973). In precise, if the graph has a (semi-)Eulerian path, that path is an optimal solution. Otherwise, we need to find the smallest number of existing edges to duplicate so that the resulting multigraph does have an (semi-)Eulerian path.

Given a (sub)graph, we can easily check whether it is (semi-)Eulerian (West et al., 2001). If it is not, we can always convert it to be a (semi-)Eulerian graph by duplicating some existing edges, which is defined as Eulerization (Edmonds & Johnson, 1973; Daubechies & Hughes, 2009). Then, we can obtain the (semi-)Eulerian path for the given graph. Since there are often more than one path for the (semi-)Eulerian graph, we randomly sample one of the paths for training. This can be regarded as data augmentation, which has been proven to be very helpful in CV (Perez & Wang, 2017).

We then re-index the nodes with $1, 2, \cdots, n$ according to the order of their appearance in the path, where $n$ is the total number of nodes. As in the example of the Fig. 1, $n$ of the path $i$ is 6, and the node $F$ is re-indexed as '1' and $A$ is re-indexed as '2'.

For graphs with node attributes, and edge weights/attributes/directions (*i.e.*, directed graphs), we represent the discrete attributes with tokens, and split the continuous numbers into individual digits as the tokens. We assign two tokens to represent the incoming and outgoing edge directions respectively. The tokens of a node's attributes are appended right after the node token (*e.g.*, '1', '2'), and the tokens of the edge's attributes are put in between the source and destination nodes (see Appendices D to F for detailed examples).

For the graph with disconnected components, we connect them by adding edges between the randomly picked nodes from different components. We use the token <edge_jump> as those edges' attributes.

We claim that the conversion of the graph to the sequence of tokens is lossless and reversible, because one can always recover the original graph by connecting the adjacent tokens. For example, in the sequence $i$ of Fig. 1, by linking '1' to '2', '2' to '3' and so on, we obtain all the edges as ensured by the Eulerian Path Theorem. With all the edges, we are able to construct a graph equivalent to the original one except for the notations of the nodes. More rigorously, they are isomorphic (Grohe & Schweitzer, 2020).

### 2.2.2 SUBGRAPH SAMPLING

Serializing a large graph as in Sec. 2.2.1 leads to an extremely long sequence, which cannot be fed into the transformer. To tackle the problem, one way is to convert the whole graph into a sequence first, and then divide it into segments that can be fed into the transformer. The other way involves sampling subgraphs and then converting them into sequences. Both Eulerizing a big graph and finding its Eulerian path are quite time-consuming. Additionally, it causes the inconsistency between the sequences in pre-training and fine-tuning. Therefore, we adopt the second way to serialize a large graph.

To be more specific, we sample a subgraph randomly around the central edge or node using the ShaDowKHop sampler introduced in Zeng et al. (2021). To ensure the resulting sequences to fit into the context window, we pre-calculate the depth and number of neighbors of the ShaDowKHop sampler. They vary across different datasets and context windows (see App. A.1 for more).

### 2.2.3 NODE IDENTITY ENCODING

When sampling subgraphs from a large graph, we need to preserve the nodes' global identity, otherwise much information will be lost. One straightforward method is to represent each node as a unique token in the transformer's vocabulary, *i.e.*, to encode each node with a learnable embedding

vector. However, when the number of nodes in the graph is huge, the vocabulary size becomes enormous, causing the model size to increase significantly, and making the training difficult.

To avoid the problem, we propose to encode each node with multiple tokens. For example, given a graph with 10 billion ($10^{10}$) nodes, we can encode each node uniquely with two tokens, and each token ranges from 1 to $10^5$. As a result, this encoding would utilize $10^5$ instead of previous $10^{10}$ tokens in the vocabulary. It can be implemented with METIS (Karypis & Kumar, 1997) to partition the graph into $10^5$ partitions, and each partition roughly has $10^5$ nodes. To further reduce the vocabulary size, we can encode each node with more tokens. For instance, encoding with five tokens results in a vocabulary size of 100. However, note that encoding each node with more tokens would increase the length of the resulting graph sequence.

The above encoding schema is in analogy with the coding of characters in computing. We can encode any characters with variable numbers of bytes in utf-8 (Allen et al., 2012, Chapter 2), and the resulting vocabulary is 256, which is the count of all possible bytes.

Through encoding each node uniquely with multiple tokens, we guarantee that the nodes' identity information is preserved during the subgraph extraction process. We show later in the ablation study in Sec. 3.5.3 that it does improve the results.

## 2.3 Modeling with the transformer decoder

In this section, we show that we can model various graph tasks with the pre-training and fine-tuning paradigm using the transformer decoder architecture on the graph sequence of tokens.

### 2.3.1 Pre-training with the NTP task

Self-supervised pre-training has been shown very beneficial for various NLP tasks (Devlin et al., 2019; Radford et al., 2019). The next-token-prediction (NTP) and masked-token-prediction tasks are two widely adopted self-supervised tasks. The NTP helps various models attain the SOTA results in many NLP tasks, and it enables the resulting model with strong generation capability (Brown et al., 2020). So we choose the NTP as our pre-training task.

As shown in the right part of Fig. 1, the NTP predicts the next node to connect in the context of graph sequence. The next node can be a new node, one of the previous nodes, or the end (*i.e.*, the <eos> token). For example, in the semi-Eulerian path $i$, given the node sequence '1', '2' and '3', we predict that the next node is a new node and assign the notation '4' to it. Subsequently, we predict to connect to the previous node '2', and so on. Finally, after we arrive at node '6', we predict the end. In the presence of node and edge attributes, the NTP also predicts the corresponding discretized tokens encoded in the graph sequence.

### 2.3.2 Fine-tuning on downstream graph tasks

As shown in Fig. 1, we append a special token <gsum> at the end of the graph sequence, and feed its embedding output to a multilayer perceptron (MLP) for the supervised graph-level task. The parameters of the transformer are initialized from the pre-trained checkpoint, while the MLP layers are randomly initialized. All the parameters are updated during the fine-tuning stage.

For the supervised edge-level task, we append the tokens of the source and destination nodes of the target edge at the end of the sequence. The output of the last token from the transformer is then fed to a MLP layer for the task. Similarly, for the node-level task, we append the target node's token at the end of the graph sequence and utilize its output for the task's MLP layer.

By reformatting the graph-/edge-/node-level tasks as above, our pre-training and fine-tuning are aligned in an elegant manifestation. The alignment allows the fine-tuning tasks to better exploit the competency of pre-training, which is verified by many NLP literature (Brown et al., 2020; Wei et al., 2021). Moreover, one can eliminate the task-specific MLP layer and convert the supervised task labels into tokens. Then the model is fine-tuned to generate these tokens, in analogy to the approach employed by GPT-3 for many NLP tasks (Brown et al., 2020).[2]

---

[2]We do not investigate it in this work, and leave it for future study.

## 3 EXPERIMENTS

### 3.1 DATASETS

Applying AI to accelerate scientific discovery has been prevailing recently (Hassabis, 2023), which motivates us to validate our proposal on the large-scale graph datasets from physical, chemical, pharmaceutical and bio-informatics domains. Also, we aim to demonstrate that our GraphGPT is applicable in the mainstream graph tasks, including graph-/edge-/node-level tasks. Therefore, we choose PCQM4Mv2 and ogbg-molpcba for the graph-level task, and ogbl-ppa for the edge-level task and ogbn-proteins for the node-level task (Hu et al., 2020; 2021). Their statistics are shown in Tab. 8 of App. A.

The PCQM4Mv2 is a quantum chemistry dataset of more than 3.7 million small organic molecules from the PubChemQC project (Nakata & Shimazaki, 2017). The ogbg-molpcba is a much small molecular dataset (Wu et al., 2017). In these 2D molecular graphs, nodes are atoms, and edges are chemical bonds. The node attributes are 9-dimensional, containing atomic number, chirality and other atom features. The edge attributes are 3-dimensional, *i.e.*, bond type, bond stereochemistry and conjugation.

In the ogbl-ppa dataset, the nodes are proteins from 58 different species, and edges represent biologically meaningful associations between proteins (Szklarczyk et al., 2019). It is a large graph of over 30 millions edges.

The ogbn-proteins dataset is an undirected, weighted, and typed dense graph. The nodes are proteins from 8 species, and the edges' 8-dimensional attributes represent 8 types of association strengths. It has nearly 40 millions edges. See more implementation details in App. C.

### 3.2 GRAPH-LEVEL TASKS

Table 1: Results of the graph classification task on the ogbg-molpcba dataset. All the baseline results are from the OGB leaderboard and the corresponding papers. The best results are in bold, and second-best are underlined.

| Models | Average Precision (AP) | | Params |
| --- | --- | --- | --- |
| | Test | Valid | |
| GCN | 0.2020±0.0024 | 0.2059±0.0033 | 0.57M |
| GIN | 0.2266±0.0028 | 0.2305±0.0027 | 1.92M |
| GINE+bot | 0.2994±0.0019 | 0.3094±0.0023 | 5.51M |
| Nested GIN+virtual node | 0.3007±0.0037 | 0.3059±0.0056 | 44.19M |
| PDF | **0.3031±0.0026** | **0.3115±0.0020** | 3.84M |
| GraphGPT-mini | 0.2385±0.0012 | 0.2777±0.0017 | 4.48M |
| GraphGPT-base | 0.2517±0.0041 | 0.2857±0.0033 | 114.12M |
| GraphGPT-large | 0.2722±0.0022 | 0.2966±0.0027 | 403.84M |

For the two molecular datasets, the tasks are to predict quantum chemical property only from 2D molecular graphs without their 3D equilibrium structures, which is practically favorable. We predict 128 binary molecular properties for ogbg-molpcba and the HOMO-LUMO gap for PCQM4Mv2.

We experiment 3 models of parameters from 4.48M to 403.84M for the ogbg-molpcba dataset. We pre-train the models with 16B tokens, and then fine-tuned for 10 epochs. The results from the epoch that maximizes the validation metric are reported in Tab. 1.

Our GraphGPT produces results much better than the powerful GNNs like GCN and GIN (Kipf & Welling, 2017; Xu et al., 2019). Our results also approach the SOTA from those sophisticated GNNs. In addition, GraphGPT yields consistently improved results as we increase the model size up to 400M. This verifies that GraphGPT does not suffer from the over-smoothing or over-squashing problems common in GNNs. It suggests that GraphGPT can perform even better if pre-trained with more data and more parameters.

Table 2: Results of the graph regression task on the PCQM4Mv2 dataset. The metric is mean absolute error (MAE), the smaller the better. We highlight the results that do not use 3D data.

| | Models | Use 3D | MAE Valid | MAE Test | Params |
|---|---|---|---|---|---|
| GNNs | GCN | ✗ | 0.1379 | 0.1398 | 2.0M |
| | GIN | ✗ | 0.1195 | 0.1218 | 3.8M |
| | GCN-VN | ✗ | 0.1153 | 0.1152 | 4.9M |
| | GIN-VN | ✗ | 0.1083 | 0.1084 | 6.7M |
| GTs | TokenGT | ✗ | 0.0910 | 0.0919 | 48.5M |
| | Graphformer | ✗ | **0.0864** | N/A | 48.3M |
| | Transformer-M | ✓ | 0.0772 | 0.0782 | 69.0M |
| | Uni-Mol+ | ✓ | 0.0693 | 0.0705 | 770.3M |
| Ours | GraphGPT | ✗ | **0.0875** | N/A | 49.7M |

As for PCQM4Mv2, we compare to two GTs that do not utilize the molecular 3D data, *i.e.*, TokenGT and Graphformer (Kim et al., 2022; Ying et al., 2021). We configure our transformer backbone the same as theirs for a fair comparison. The resulting model has 49.7M parameters, and is pre-trained with about 49.5B tokens.

The TokenGT uses the vanilla transformer backbone, and it serializes the graph into tokens by directly unfolding all the nodes and edges. The serialization method prohibits it from being self-supervised pre-trained. TokenGT also supplements the inputs with handcrafted structural features.

The Graphformer transforms the graph into the sequence of tokens using nodes only. The structural information possessed in the edges are completely lost in the input sequence, so Graphformer has to add them back through modifying the transformer's attention layers and enhancing the inputs with sophisticated tricks like centrality encoding and spatial encoding to handcraft structural information. Same as TokenGT, Graphformer cannot adopt self-supervised pre-training.

In contrast, our GraphGPT does not require any handcrafted features, and obtains better results than TokenGT on the valid dataset, *i.e.*, 0.0875 versus 0.0910. Our result is comparable to Graphformer. What's more, GraphGPT surpasses all the GNNs by a large margin. The exceptional performance of GraphGPT implies that our lossless serialization and the self-supervised generative pre-training enable the model to fully capture the graph's structural and semantic information. We are pre-training larger GraphGPT with more compute budget in order to attain better results.

## 3.3 EDGE-LEVEL TASKS

The edge-level task of the dataset ogbl-ppa is link prediction. We use two tokens for the node identity encoding introduced in Sec. 2.2.3. Specifically, we use the species to partition the nodes, so the first token represents the species, and the second is the local indices of proteins inside each species (see App. E for details).

We experiment 5 different model sizes ranging from 14.75M to 444.92M, and report 3 of them in Tab. 3 (see App. E for more). GraphGPT-mini/base/large are pre-trained with 25.6B/39B/50B tokens respectively, and then fine-tuned with the classification task. The fine-tuning data consists of subgraphs induced by the positive edges for training and equal negative edges randomly sampled.

As in Tab. 3, GraphGPT performs much better than all kinds of models, including GNNs, Heuristic and Latent Factor models. Our smallest model outperforms the strong GNNs like GCN, GraphSAGE and SEAL a lot (Kipf & Welling, 2017; Hamilton et al., 2017; Zhang & Chen, 2018). Our best result 67.15 surpasses the current SOTA 63.22 by a large margin.

AGDN (Sun et al., 2020), the largest GNN in the leaderboard, has only 36.9M parameters. The prevailing GNNs like GCN, GraphSAGE and SEAL have less than 1M parameters. They cannot be further scaled up to achieve better results due to the over-smoothing and over-squashing problems.

In contrast, GraphGPT is able to scale up to 400M+ with consistently increasing performance. This motives further study on larger models and more data with more compute budget.

What's more, previously no GTs are listed in the ogbl-ppa's leaderboard, and we show that transformer based models can excel in edge-level tasks.

Table 3: Results of the link prediction task on the ogbl-ppa dataset.

| Models | | HR@100 (%) | | Params |
| | | Test | Valid | |
| --- | --- | --- | --- | --- |
| Heuristic | Common Neighbor | 27.65±0.00 | 28.23±0.00 | 0 |
| | Adamic Adar | 32.45±0.00 | 32.68±0.00 | 0 |
| | Resource Allocation | 49.33±0.00 | 47.22±0.00 | 0 |
| Latent Factor | DeepWalk | 28.88±1.53 | - | 150.14M |
| | Matrix Factorization | 32.29±0.94 | 32.28±4.28 | 147.66M |
| GNN | GCN | 18.67±1.32 | 18.45±1.40 | 0.28M |
| | GraphSAGE | 16.55±2.40 | 17.24±2.64 | 0.42M |
| | SEAL | 48.80±3.16 | 51.25±2.52 | 0.71M |
| | AGDN | 41.23±1.59 | 43.32±0.92 | 36.90M |
| | SIEG | 63.22±1.74 | 65.33±2.34 | 1.99M |
| GraphGPT (Ours) | mini | 55.56±1.14 | 54.87±0.66 | 14.75M |
| | base | 64.98±1.73 | 66.68±1.33 | 144.93M |
| | large | **67.15±1.36** | **68.60±1.40** | 444.92M |

## 3.4 NODE-LEVEL TASKS

The task of the ogbn-proteins dataset is to predict 112 binary labels of proteins that each indicates the presence of one type of function. The node identity is encoded with two tokens similar to the ogbl-ppa in Sec. 3.3 (see App. F for details).

We pre-train 5 different models that scale from 10.76M to 428.94M with 51.2B tokens, and then fine-tune them for 16 epochs. We pick the 3 consecutive epochs that maximize the metric of valid data and then calculate the mean and variance. We report the results of 3 models in Tab. 4.

For this dataset, GNNs usually use random partition (Li et al., 2020) to sample subgraphs. The resulting subgraphs usually have more than 20,000 nodes and 1 million edges. In contrast, our subgraphs usually have about 10 nodes and 20 edges, because we limit the transformers' context window to be 256. It is remarkable that GraphGPT's results surpass GCN and GraphSAGE, and approach the SOTA with such a small neighborhood to gather information. We hypothesize that during the pre-training, the global structural and semantic information of the big graph have been encoded in the node tokens' embeddings and the transformer's parameters.

GNNs in the leaderboard are very small. For example, GCN and GraphSAGE has 0.1M to 0.2M parameters, and even DeeperGCN does not exceed 2.37M (Li et al., 2020). The largest GNN model RevGNN-wide has 68M parameters (Li et al., 2021). In contrast, our GraphGPT-large has 400M+ parameters and can be well trained. Compared to the only transformer based model UniMP in the leaderboard (Shi et al., 2020), GraphGPT is able to train much larger transformers and attains improved results.

Despite of the fine performance of GraphGPT, the small context window may restrain its capability. Techniques like Flash Attention (Dao et al., 2022) and Positional Interpolation (Chen et al., 2023) can be utilized to make the context window much longer.[3]

---

[3]We experimented with the context window of 1024, and observed some gains.

Table 4: Results of the node classification task on the ogbn-proteins dataset.

| Models | ROC-AUC (%) Test | Valid | Params |
|---|---|---|---|
| GCN | 72.51±0.35 | 79.21±0.18 | 0.10M |
| GraphSAGE | 77.68±0.20 | 83.34±0.13 | 0.19M |
| DeeperGCN | 85.80±0.17 | 91.06±0.16 | 2.37M |
| UniMP | 86.42±0.08 | 91.75±0.06 | 1.91M |
| RevGNN-wide | 88.24±0.15 | 94.50±0.08 | 68.47M |
| GIPA (Wide&Deep) | **89.17±0.07** | **94.72±0.20** | 17.44M |
| GraphGPT-mini | 75.61±1.37 | 80.47±0.94 | 10.76M |
| GraphGPT-base | 83.37±0.15 | 87.68±0.25 | 132.94M |
| GraphGPT-large | 84.80±0.18 | 89.35±0.24 | 428.94M |

## 3.5 ABLATION STUDY

We present the ablation study on three key ingredients of GraphGPT, *i.e.*, pre-training, node re-indexing and node identity encoding.

### 3.5.1 PRE-TRAINING

Pre-training with the NTP task is the key to the success of our GraphGPT. Experiment results on graph/edge/node-level tasks in Tab. 5 show that pre-training can boost the performance a lot. The lift of metrics ranges from 30% to 100%. This implies that pre-training enables the model to understand both the graph structure and the semantics possessed in the node and edge attributes.

Table 5: Ablation study of pre-training on the datasets of various types of tasks.

| Task Type | Datasets | Params | Metrics | Pre-training | Test | Valid |
|---|---|---|---|---|---|---|
| Graph | ogbg-molpcba | 4.48M | AP | ✗ | 0.1280 | 0.1331 |
| | | | | ✓ | **0.2385** | **0.2777** |
| | PCQM4Mv2 | 49.7M | MAE | ✗ | N/A | 0.1086 |
| | | | | ✓ | N/A | **0.0875** |
| Edge | ogbl-ppa | 14.75M | HR@100 | ✗ | 41.28 | 40.14 |
| | | | | ✓ | **55.56** | **54.87** |
| Node | ogbn-proteins | 10.76M | ROC-AUC | ✗ | 57.52 | 61.19 |
| | | | | ✓ | **75.61** | **80.47** |

### 3.5.2 NODE RE-INDEXING

As shown in Fig. 1, we re-index the nodes according to their order in the (semi-)Eulerian path. We conduct experiments using the ogbg-molpcba dataset to assess its effectiveness. As shown in Tab. 6, node re-indexing improves the downstream tasks for different model sizes, although it increases the pre-training loss. This is because node re-indexing can be regarded as data augmentation. It prevents the model from memorizing the unimportant information of the graph, *i.e.*, the notation of the nodes, and leads to better generalization performance. In addition, the re-indexing allows us to restrict the decoding space of node tokens when generating graphs with GraphGPT.

### 3.5.3 NODE IDENTITY ENCODING

As illustrated in Sec. 2.2.3, we encode each node in a big graph uniquely with multiple tokens. Here we show its importance for edge-/node-level tasks. We use GraphGPT-mini to conduct the ablation

Table 6: Ablation study of node re-indexing on the ogbg-molpcba dataset with two model sizes.

| Params | Metrics | Re-index | Pre-training Loss | Test | Valid |
|---|---|---|---|---|---|
| 114.12M | AP | ✗ | **0.0689** | 0.2270 | 0.2621 |
| | | ✓ | 0.0750 | **0.2517** | **0.2857** |
| 4.48M | AP | ✗ | **0.0844** | 0.2310 | 0.2525 |
| | | ✓ | 0.0874 | **0.2385** | **0.2777** |

study to save time and compute budget. Node identity encoding can obviously improve the results as in Tab. 7. For more details, see appendices A and E.

Table 7: Ablation study of node identity encoding on the ogbl-ppa and ogbn-proteins datasets.

| Task Type | Datasets | Params | Metrics | Node ID encoding | Test | Valid |
|---|---|---|---|---|---|---|
| Edge | ogbl-ppa | 14.75M | HR@100 | ✗ | 44.38 | 45.08 |
| | | | | ✓ | **55.56** | **54.87** |
| Node | ogbn-proteins | 10.76M | ROC-AUC | ✗ | 60.22 | 65.66 |
| | | | | ✓ | **75.61** | **80.47** |

## 4 RELATED WORKS

***Graph Neural Networks (GNNs)*** GNNs have been dominating the graph learning in the past decades. They have many variants, and gain excellent performance in various graph tasks (Wu et al., 2020). However, most GNNs suffer from the over-smoothing and over-squashing problems, which restrict them from scaling up (Rusch et al., 2023; Alon & Yahav, 2021).

***Graph Transformers (GTs)*** Inspired by the success of transformers in NLP and CV, graph community starts to embrace transformer architectures in recent years (Ying et al., 2021; Rampásek et al., 2022; Müller et al., 2023). The various GTs have attained some good results, especially in graph-level tasks with large-scale datasets (Müller et al., 2023). Nevertheless, they have to employ handcrafted features or GNN modules to encode the structure information in the inputs (Ying et al., 2021; Kim et al., 2022), and/or in the self-attention layers (Ying et al., 2021; Chen et al., 2022; Luo et al., 2023). In addition, they mostly adopt the transformer encoder instead of the decoder, preventing them from being pre-trained in a self-supervised manner.

***Pre-training and fine-tuning*** After the invention of the transformer (Vaswani et al., 2017), the self-supervised pre-training and supervised fine-tuning paradigm starts to flourish in NLP and brings significant improvements across various tasks (Devlin et al., 2019; Radford et al., 2018). Pre-training with web-scale text data (Brown et al., 2020) and followed by instruction tuning (Wei et al., 2021) or reinforcement learning from human feedback (Ouyang et al., 2022) advances the paradigm and boosts the performance even further. In CV, pre-training with the large scale supervised datasets and fine-tuning on the small datasets gains much success, which is termed as transfer learning (Yosinski et al., 2014). Recently, MAE from He et al. (2022) shows that self-supervised pre-training by predicting large portion of masked image patches can also attain SOTA results.

## 5 CONCLUSIONS

We propose the novel model GraphGPT that can be adapted for the graph-/edge-/node-level tasks, and it can attain SOTA or close to SOTA results. With GraphGPT, we train large models of up to hundreds of millions parameters, and observe consistent performance increase. GraphGPT has the potential to be scaled up to hundreds of billions parameters, and can be aligned or integrated with large language models. For a comprehensive understanding of GraphGPT, we discuss its limitations in App. G.

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

## A  DATASETS

The detailed statistics of datasets are reported in Tab. 8.

Table 8: Statistics of graph-/edge-/node-level datasets. Here 'BC' stands for binary classification.

| datasets | # of graphs | avg # of nodes | avg # of edges | task-type | metrics |
|---|---|---|---|---|---|
| ogbg-molpcba | 437,929 | 26.0 | 28.1 | BC | AP |
| PCQM4Mv2 | 3,746,619 | 14.14 | 14.56 | regression | MAE |
| ogbl-ppa | 1 | 576,289 | 30,326,273 | multi-label BC | HR@100 |
| ogbn-proteins | 1 | 132,534 | 39,561,252 | BC | ROC-AUC |

### A.1  SUBGRAPH SAMPLING

The subgraph sampling for different datasets of big graphs are shown in Tab. 9.

Table 9: Details of subgraph sampling for ogbl-ppa and ogbn-proteins datasets. 'ctx' means the context window. edge-ego means sampling subgraph around the central edge, and node-ego means around the central node.

| dataset | sampling | depth | # neighbors | ctx |
|---|---|---|---|---|
| ogbl-ppa | edge-ego | 1 | 14 | 256 |
| ogbn-proteins | node-ego | 9 | 1 | 256 |
| | | 16 | 1 | 512 |
| | | 24 | 1 | 1024 |
| | | 42 | 1 | 2048 |

## B  MODELS

We list the model specifics in Tab. 10.

Table 10: Statistics of GraphGPT models of different sizes. The GraphGPT-base is of the same scale as Bert-base (Devlin et al., 2019).

| model-size | hidden-size | # of layers | # of heads |
|---|---|---|---|
| mini | 256 | 4 | 4 |
| small | 512 | 4 | 8 |
| medium | 512 | 8 | 8 |
| base | 768 | 12 | 12 |
| large | 1024 | 24 | 16 |

## C  IMPLEMENTATION DETAILS

### C.1  GRAPHS TO SEQUENCES OF TOKENS

The code is implemented with Pytorch. We use torch-geometric (Fey & Lenssen, 2019) to preprocess graphs, *e.g.*, subgraph sampling and etc. We use Networkx (Hagberg et al., 2008) to Eulerize the (sub)graphs if needed and then find the (semi-)Eulerian paths. We write our own tokenizer to convert the (semi-)Eulerian paths to sequences of tokens. The token vocabulary is built for each dataset separately.

### C.2  MODEL BACKBONE

We adopt Llama's transformer decoder (Touvron et al., 2023) implemented in the Hugging Face's transformers package (Wolf et al., 2020) as our backbone. We do not used Llama's pre-trained weights. Instead, we train models of different scales as in Tab. 10 with random parameters initialization.

### C.3  TRAINING

The models are pre-trained and fine-tuned on GPU clusters of V100-32G using DeepSpeed's stage-2 schema with mixed precision (Rasley et al., 2020). Following the optimizer setting in Llama, we use AdamW (Loshchilov & Hutter, 2017) with the hyper-parameters $\beta_1 = 0.9$, $\beta_2 = 0.95$ and weight decay of 0.1 and gradient clipping of 1.0. We use a linear decay learning rate schedule with 1,000 warmup for pre-training only. The maximal learning rate in is $3 \times 10^{-4}$ for pre-training and $3 \times 10^{-5}$ for fine-tuning. To make full use of computing power, we pack several graph sequences together into one entry to maximize the context window (Raffel et al., 2020).

## C.4 Vocabulary

In NLP, the vocabulary is usually built by tokenizing the text data with the byte-pair encoding (BPE) algorithm (Sennrich et al., 2016). The resulting unique tokens form the vocabulary, and they are usually frequent subwords in the text corpus.

In our GraphGPT, the vocabulary is constructed very differently. We split the vocabulary into two parts, the first part contains the structural and special tokens, and they are dataset agnostic. The second part consists of the tokens that encode the semantics information of the dataset, such as node and edge attributes.

App. E shows an example. In the graph sequence, tokens '1', '2' and so on are structural tokens. 'ogbl-ppa#node#0#17' and 'ogbl-ppa#node#1#1959' are semantics tokens. $<$gsum$>$ and $<$eos$>$ in Fig. 1 are special tokens.

## D Graph-level task

### D.1 Graphs to sequences of tokens

Below is one example of 2D molecular graphs in the ogbg-molpcba dataset in torch-geometric data format (Fey & Lenssen, 2019).

```
Data(x=[4, 9], edge_index=[2, 6], edge_attr=[6, 3], y=[128])
```

The graph has 4 nodes and 3 edges. The source and destination nodes of the edges are recorded in 'edge_index', and its dimension is $(2, 2 \cdot \text{number\_of\_nodes})$ for undirected graphs. 'x' is the node attributes of 9 dimensions, and 'edge_attr' stores the edge attributes of 3 dimensions.

The node and edge attributes of the graphs are numbers. If we directly discretize them into tokens, i.e., using one token to represent each unique number, the numbers that appear few times in the dataset cannot be well-trained. At the same time, the vocabulary may blow up. Therefore, we split them into single digits and represent them with the combination of the following tokens. They are dataset agnostic, and can be shared across different datasets.

$<->$, $<.>$, $<0>$, $<1>$, $<2>$, $<3>$, $<4>$, $<5>$, $<6>$, $<7>$, $<8>$, $<9>$

The resulting vocabulary is 556 for both ogbg-molpcba and PCQM4Mv2.

Below shows the tokens from one of the possible (semi-)Eulerain paths of the above molecular graph.

```
['1', 'ogbg-molpcba#node#0#1', '<7>', 'ogbg-molpcba#node#2#1',
  '<1>', 'ogbg-molpcba#node#3#1', '<5>', 'ogbg-molpcba#node
  #6#1', '<1>', 'ogbg-molpcba#edge#0#1', '<1>', '2', '3', 'ogbg-
  molpcba#node#0#1', '<5>', 'ogbg-molpcba#node#2#1', '<4>', '
  ogbg-molpcba#node#3#1', '<5>', 'ogbg-molpcba#node#4#1', '<3>',
   'ogbg-molpcba#node#6#1', '<2>', '2', 'ogbg-molpcba#node#0#1',
   '<5>', 'ogbg-molpcba#node#2#1', '<3>', 'ogbg-molpcba#node
  #3#1', '<5>', 'ogbg-molpcba#node#6#1', '<1>', '4', 'ogbg-
  molpcba#node#0#1', '<5>', 'ogbg-molpcba#node#2#1', '<4>', '
  ogbg-molpcba#node#3#1', '<5>', 'ogbg-molpcba#node#4#1', '<3>',
   'ogbg-molpcba#node#6#1', '<2>']
```

In the sequence of tokens above, for the node '1', we can deduce that its 9 dimensional attributes are $(7, 0, 1, 5, 0, 0, 1, 0, 0, 0)$. Node '1' is connected to '2' with edge attributes $(1, 0, 0)$. We set 0 as the the default value of the attributes in this dataset, and do not encode it into tokens.

In the (semi-)Eulerian path, a node may appear several times. We append its attributes tokens to one of its appearances randomly. This can prevent the model from copying the attributes from the previous appearance, and also shorten the resulting sequence.

For a graph obtained from Eulerization, an edge may present several times in the path. We apply the same logic to insert the edge attributes tokens.

As in the above sequence, node '2' appears two times, and its node attributes tokens are appended after its second appearance. There is no tokens encode the edge attributes of edge between '2' and '3', which implies the edge attributes are default value $(0, 0, 0)$.

In the ablation study on node re-index in Sec. 3.5.2, the resulting tokens without node re-indexing of the same example above is shown below. Without re-indexing, the starting node will keep its original notation, which is '4' in this case.

```
['4', 'ogbg−molpcba#node#0#1', '<7>', 'ogbg−molpcba#node#2#1',
   '<1>', 'ogbg−molpcba#node#3#1', '<5>', 'ogbg−molpcba#node
   #6#1', '<1>', 'ogbg−molpcba#edge#0#1', '<1>', '2', '1', 'ogbg−
   molpcba#node#0#1', '<5>', 'ogbg−molpcba#node#2#1', '<4>', '
   ogbg−molpcba#node#3#1', '<5>', 'ogbg−molpcba#node#4#1', '<3>',
   'ogbg−molpcba#node#6#1', '<2>', '2', 'ogbg−molpcba#node#0#1',
   '<5>', 'ogbg−molpcba#node#2#1', '<3>', 'ogbg−molpcba#node
   #3#1', '<5>', 'ogbg−molpcba#node#6#1', '<1>', '3', 'ogbg−
   molpcba#node#0#1', '<5>', 'ogbg−molpcba#node#2#1', '<4>', '
   ogbg−molpcba#node#3#1', '<5>', 'ogbg−molpcba#node#4#1', '<3>',
   'ogbg−molpcba#node#6#1', '<2>']
```

## D.2 MODEL, PRE-TRAINING AND FINE-TUNING

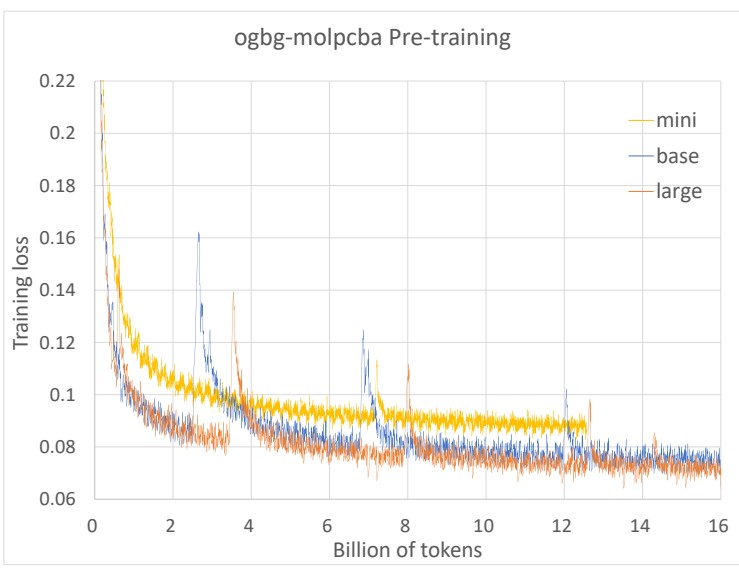

Figure 2: Training loss versus tokens of ogbg-molpcba dataset for models small/base/large as in Tab. 10. They are all trained on 12.6 B to 16 B tokens with a batch size of 0.4M tokens.

We set the context window to be 1024 so that the token sequence of all the molecules can be fit in. We use the mini-batch of 1024 and 1920 sequences for the ogbg-molpcba and PYQM4Mv2 datasets respectively. The total update-step is $4.3 \times 10^4$ and $1.8 \times 10^6$.

## E EDGE-LEVEL TASK

### E.1 GRAPHS TO SEQUENCES OF TOKENS

The whole ogbl-ppa dataset is summarized in torch-geometric format as follows.

```
Data(num_nodes=576289, edge_index=[2, 42463862], x=[576289, 58])
```

It has 576289 nodes and 21231931 edges in the training data. 'x' is the one-hot representation of the species that the node (protein) belongs to.

We sample a subgraph from it as below.

```
Data(num_nodes=30, root_n_id=[2], edge_index=[2, 84], x=[30, 2])
```

It has 30 nodes, 42 edges as in 'edge_index'. 'x' is the node attributes of 2 dimensions, and it encodes the node identity as described in Sec. 2.2.3. We partition the nodes (proteins) based on the associated species. The number of proteins inside each species varies from 616 to 41017. Finally we use 58 tokens for species and 41017 tokens for the local indices. Combined with the tokens for the structure and the special tokens, the total vocabulary is 41231.

Here 'root_n_id' records the two seed nodes, and the subgraph is sampled centered around them. The resulting tokens from one of the possible (semi-)Eulerian paths are:

```
['1', '2', '3', 'ogbl-ppa#node#0#17', 'ogbl-ppa#node#1#1959', '4',
    '5', 'ogbl-ppa#node#0#17', 'ogbl-ppa#node#1#2460', '6', '7',
    'ogbl-ppa#node#0#17', 'ogbl-ppa#node#1#3566', '6', '8', 'ogbl-
    ppa#node#0#17', 'ogbl-ppa#node#1#4145', '6', '9', 'ogbl-ppa#
    node#0#20', 'ogbl-ppa#node#1#5334', '10', 'ogbl-ppa#node
    #0#27', 'ogbl-ppa#node#1#17324', '6', 'ogbl-ppa#node#0#17', '
    ogbl-ppa#node#1#6850', '11', 'ogbl-ppa#node#0#17', 'ogbl-ppa#
    node#1#5498', '6', '12', 'ogbl-ppa#node#0#17', 'ogbl-ppa#node
    #1#5776', '6', '4', 'ogbl-ppa#node#0#17', 'ogbl-ppa#node
    #1#8183', '2', '5', '2', '13', 'ogbl-ppa#node#0#17', 'ogbl-ppa
    #node#1#3514', '2', 'ogbl-ppa#node#0#17', 'ogbl-ppa#node
    #1#9374', '14', 'ogbl-ppa#node#0#17', 'ogbl-ppa#node#1#6164',
    '15', 'ogbl-ppa#node#0#17', 'ogbl-ppa#node#1#8368', '2', '6',
    '16', 'ogbl-ppa#node#0#17', 'ogbl-ppa#node#1#10803', '6',
    '17', 'ogbl-ppa#node#0#17', 'ogbl-ppa#node#1#11465', '6',
    '10', '18', 'ogbl-ppa#node#0#20', 'ogbl-ppa#node#1#16505',
    '6', '19', 'ogbl-ppa#node#0#17', 'ogbl-ppa#node#1#15071', '2',
    '20', 'ogbl-ppa#node#0#17', 'ogbl-ppa#node#1#7761', '2',
    '21', 'ogbl-ppa#node#0#17', 'ogbl-ppa#node#1#8828', '2', '22',
    'ogbl-ppa#node#0#17', 'ogbl-ppa#node#1#14477', '2', '23', '
    ogbl-ppa#node#0#17', 'ogbl-ppa#node#1#16026', '2', '24', 'ogbl
    -ppa#node#0#17', 'ogbl-ppa#node#1#16825', '6', '25', 'ogbl-ppa
    #node#0#17', 'ogbl-ppa#node#1#17615', '19', '25', '2', '26', '
    ogbl-ppa#node#0#17', 'ogbl-ppa#node#1#19524', '2', '27', 'ogbl
    -ppa#node#0#17', 'ogbl-ppa#node#1#17854', '6', '28', 'ogbl-ppa
    #node#0#17', 'ogbl-ppa#node#1#17733', '6', '29', 'ogbl-ppa#
    node#0#27', 'ogbl-ppa#node#1#23255', '6', '30', 'ogbl-ppa#node
    #0#17', 'ogbl-ppa#node#1#19700', '6', '27', '1', 'ogbl-ppa#
    node#0#17', 'ogbl-ppa#node#1#20474']
```

In the ablation study on node identity encoding in Sec. 3.5.3, an example of the subgraph sampled from ogbl-ppa without identity encoding is shown below.

```
Data(num_nodes=30, root_n_id=[2], edge_index=[2, 136], x=[30, 1])
```

Different from the subgraph with node identity encoded in 'x', its node attribute 'x' contains only the information of the node's (protein) hosting species. It cannot be used to uniquely identify the nodes. The vocabulary decreases from 41231 to 214.

The resulting tokens from one of its possible (semi-)Eulerian paths is below.

```
['1', '2', '3', '4', 'ogbl-ppa#node#0#17', '5', '6', '7', '5',
    '8', '9', '1', 'ogbl-ppa#node#0#17', '10', 'ogbl-ppa#node
    #0#17', '11', 'ogbl-ppa#node#0#17', '3', 'ogbl-ppa#node#0#17',
    '11', '12', '1', '5', '13', 'ogbl-ppa#node#0#17', '5', '14',
    'ogbl-ppa#node#0#17', '5', '9', '10', '8', 'ogbl-ppa#node
```

```
#0#17', '3', '15', 'ogbl-ppa#node#0#17', '3', '16', 'ogbl-ppa#
node#0#17', '3', '2', 'ogbl-ppa#node#0#20', '17', 'ogbl-ppa#
node#0#27', '1', '18', 'ogbl-ppa#node#0#20', '1', '19', 'ogbl-
ppa#node#0#17', '3', '9', 'ogbl-ppa#node#0#17', '20', 'ogbl-
ppa#node#0#17', '10', '3', '21', '3', '5', '10', '12', 'ogbl-
ppa#node#0#17', '3', '22', 'ogbl-ppa#node#0#17', '3', '17',
'18', '3', '23', '13', '24', '5', '25', 'ogbl-ppa#node#0#17',
'23', 'ogbl-ppa#node#0#17', '21', 'ogbl-ppa#node#0#17', '20',
'5', '26', 'ogbl-ppa#node#0#17', '5', '22', '24', 'ogbl-ppa#
node#0#17', '23', '5', '27', '6', 'ogbl-ppa#node#0#17', '28',
'ogbl-ppa#node#0#17', '7', 'ogbl-ppa#node#0#17', '28', '5', '
ogbl-ppa#node#0#17', '27', 'ogbl-ppa#node#0#17', '29', 'ogbl-
ppa#node#0#17', '5', '30', 'ogbl-ppa#node#0#17', '5', '19',
'5', '12', '20', '1']
```

## E.2 MODEL, PRE-TRAINING AND FINE-TUNING

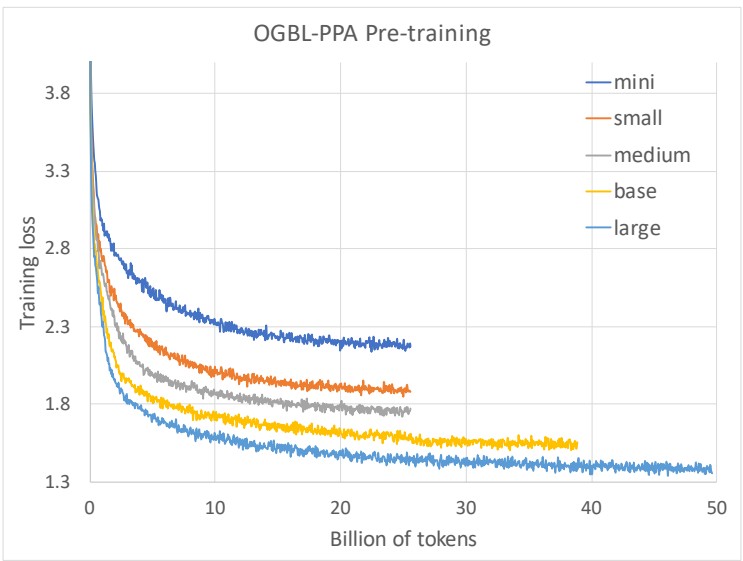

Figure 3: Training loss versus tokens of ogbl-ppa dataset for models mini/small/medium/base/large as in Tab. 10. They are all trained on 25.6 B tokens with a batch size of 0.4M (*i.e.*, $1600 \times 256$) tokens. GraphGPT-base and GraphGPT-large are further trained with additional tokens to attain better fine-tuning performance.

In pre-training stage, we use the mini-batch of 1600 sequences of 256 tokens. The total update-step is $6.25 \times 10^4$. In fine-tuning stage, we use the mini-batch of 4096 sequences of maximal 256 tokens.

The pre-training loss versus the number of tokens is shown in Fig. 3. In general, larger model results in lower pre-training loss, and better results in down-stream fine-tuning tasks. The fine-tuning results of link prediction is shown in Tab. 11. Pre-training can improve the downstream task substantially (see Fig. 4).

## E.3 NODE IDENTITY ENCODING

The comparison between models trained with and without node identity encoding is shown in Fig. 5. The node identity encoding can achieve much better results.

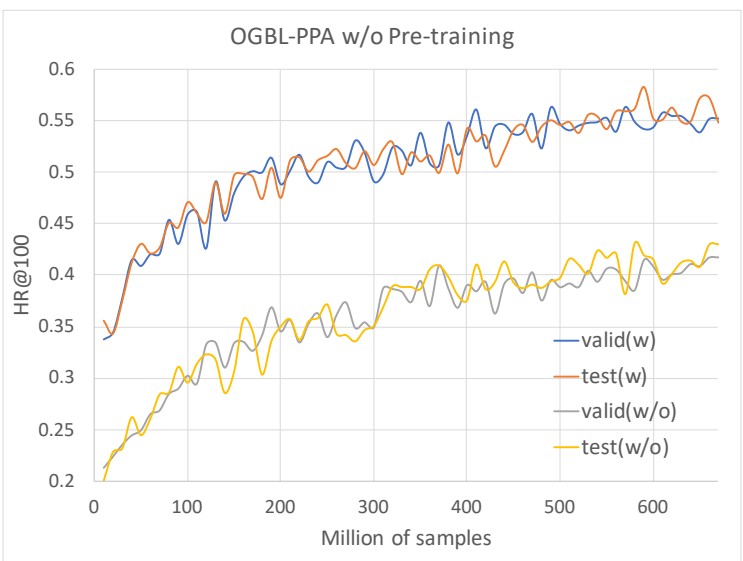

Figure 4: The ablation study of pre-training for ogbl-ppa. Evaluation metric HR@100 on the valid/test data versus the number of samples fine-tuned. 'w' and 'w/o' indicate whether pre-training is employed or not. 'w' means the model is pre-trained with 25.6 B tokens first, and then fine-tuned with the link prediction task. 'w/o' indicates that the model is trained with the supervised link prediction task directly with random parameters initialization. We use GraphGPT-mini to save time and computing resources.

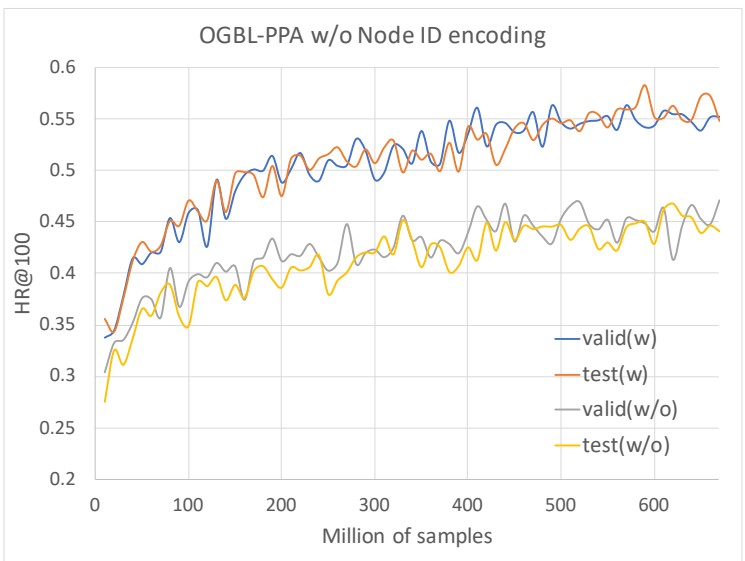

Figure 5: The ablation study of node identity encoding for ogbl-ppa. Evaluation metric HR@100 on the valid/test data versus the number of samples fine-tuned. 'w' and 'w/o' indicate whether the node is encoded with two tokens uniquely or not. Both 'w' and 'w/o' are pre-trained with 25.6 B tokens first, and then fine-tuned with the link prediction task. After fine-tuning with about 500 M samples, the results become stable. The model is GraphGPT-mini.

Table 11: Edge-level link prediction task results of the ogbl-ppa dataset of different model size.

| Models | | HR@100 (%) | | Params |
|--------|--|------------|--|--------|
| | | Test | Valid | |
| GraphGPT | mini (w/o pre-training) | 41.28±1.35 | 40.14±1.01 | 14.75M |
| | mini | 55.56±1.14 | 54.87±0.66 | 14.75M |
| | small | 57.25±1.34 | 58.78±0.68 | 37.89M |
| | medium | 60.08±1.18 | 61.01±0.77 | 54.67M |
| | base | 64.98±1.73 | 66.68±1.33 | 144.93M |
| | large | 67.15±1.36 | 68.60±1.40 | 444.92M |

### E.4 OTHER DATASETS

We also conduct the experiment on a small dataset ogbl-ddi. The performance is not good. It either implies that we need more experiments to find suitable hyper-parameters, or indicates that GraphGPT might not be a good choice for small dense graphs. ogbl-ddi has around 14.67% edges of the corresponding complete graph. In contrast, ogbl-ppa has only 0.018% edges of the complete graph.

## F NODE-LEVEL TASK

### F.1 GRAPHS TO SEQUENCES OF TOKENS

The entire ogbn-proteins dataset is a large graph as follows.

```
Data(num_nodes=132534, edge_index=[2, 79122504], edge_attr
    =[79122504, 8], node_species=[132534, 1], y=[132534, 112])
```

It has 132,534 nodes and 39,561,252 edges. 'node_species' stores the species' numeric id that the node (proteins) belongs to.

One sampled subgraph in the torch-geometric data format is:

```
Data(num_nodes=10, root_n_id=0, edge_index=[2, 22], edge_attr=[22,
    8], y=[10, 112], x=[10, 2])
```

It has 10 nodes, 11 edges as in 'edge_index'. Edge attributes is stored in 'edge_attr' of dimension 8. 'x' is the node attributes of 2 dimensions, and it encodes the node identity as described in Sec. 2.2.3. Its first dimension (token) represents the species, and the second is local numbering of each protein inside its species. Similar to the ogbl-ppa dataset, the identity encoding of 132,534 nodes occupies 25,465 tokens in the vocabulary, and the total vocabulary is 25,620.

'y' records the labels for the supervised node-level task. 'root_n_id' represents the target node, and the subgraph is sampled centered around it.

The resulting tokens from one of the possible (semi-)Eulerian paths are as follows.

```
['1', 'ogbn−proteins#node#0#3702', 'ogbn−proteins#node#1#16267', '
    ogbn−proteins#edge#7#1', '<1>', '<6>', '<4>', '2', 'ogbn−
    proteins#node#0#3702', 'ogbn−proteins#node#1#6896', 'ogbn−
    proteins#edge#4#1', '<3>', '<4>', '<0>', '3', 'ogbn−proteins#
    node#0#3702', 'ogbn−proteins#node#1#4121', 'ogbn−proteins#edge
    #4#1', '<3>', '<9>', '<8>', '4', 'ogbn−proteins#node#0#3702',
    'ogbn−proteins#node#1#3963', 'ogbn−proteins#edge#4#1', '<1>',
    '<5>', '<3>', '5', 'ogbn−proteins#node#0#3702', 'ogbn−proteins
    #node#1#8259', 'ogbn−proteins#edge#4#1', '<4>', '<8>', 'ogbn−
    proteins#edge#7#1', '<2>', '<1>', '<5>', '6', '7', 'ogbn−
    proteins#edge#7#1', '<4>', '<1>', '<8>', '8', 'ogbn−proteins#
    node#0#3702', 'ogbn−proteins#node#1#1', '7', 'ogbn−proteins#
```

```
node#0#3702', 'ogbn−proteins#node#1#89', 'ogbn−proteins#edge
#7#1', '<3>', '<2>', '<1>', '6', 'ogbn−proteins#node#0#3702',
'ogbn−proteins#node#1#955', 'ogbn−proteins#edge#7#1', '<2>',
'<7>', '<0>', '9', 'ogbn−proteins#node#0#3702', 'ogbn−proteins
#node#1#7055', 'ogbn−proteins#edge#4#1', '<1>', '<6>', '<5>',
'10', 'ogbn−proteins#node#0#3702', 'ogbn−proteins#node
#1#10010', 'ogbn−proteins#edge#4#1', '<1>', '<6>', '<9>', '4',
'5', 'ogbn−proteins#edge#4#1', '<2>', '<0>', '<7>', '3']
```

The original edge attributes are 8-dimensional vector of 3 decimal numbers from 0.001 to 1. We split them into single digits and represent them with the combination of the digits tokens as in App. D.

To reduce the number of tokens in the resultant sequences further, we multiply the number with 1000 and then minus it by 1. So we do not need to encode '.' any more. At the same time, we treat the value 0.001 (0 after the above transformation) as the default value and do not encode it with tokens.

### F.2  MODEL, PRE-TRAINING AND FINE-TUNING

In the pre-training, we use the mini-batch of 2048 sequences of 256 tokens. The total update-step is $9.76 \times 10^4$. In the fine-tuning, we use the mini-batch of 128 sequences of maximal 256 tokens.

The pre-training loss versus the number of tokens for various model sizes is shown in Fig. 6. In general, larger model results in lower training loss, and better results in down-stream fine-tuning tasks as in Tab. 12.

The loss of mini/small/medium models are almost saturated after training with 25.6 B tokens. In contrast, base/large models can be further pre-trained as the loss keeps decreasing.

Fig. 8 shows that pre-training can improve the results significantly.

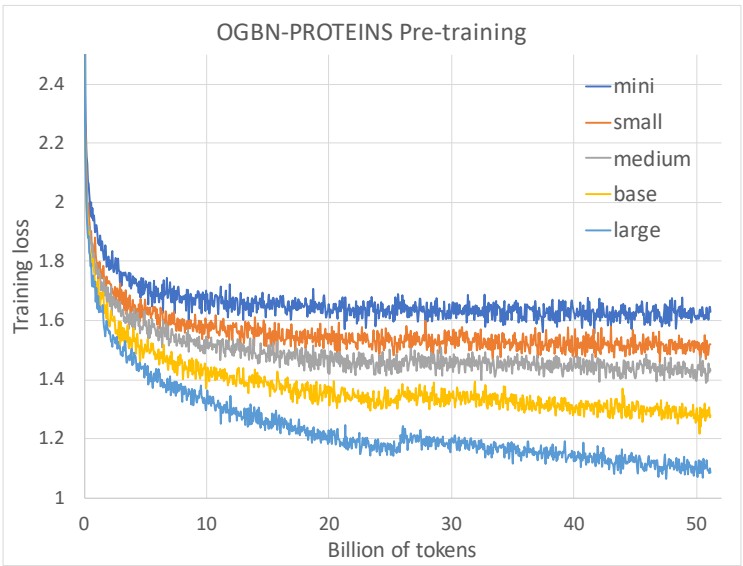

Figure 6: Pre-training loss versus tokens of ogbn-proteins dataset for models mini/small/medium/base/large as in Tab. 10. They are all trained on 51.2 B tokens with a batch size of 0.52M (*i.e.*, $2048 \times 256$) tokens. The first 25.6 B tokens are trained with the maximal learning rate $3 \times 10^{-4}$, and the second 25.6 B tokens are trained with the maxial learning rate $1 \times 10^{-4}$. The learning rate scheduler is linear decay with 1000 warm-up steps.

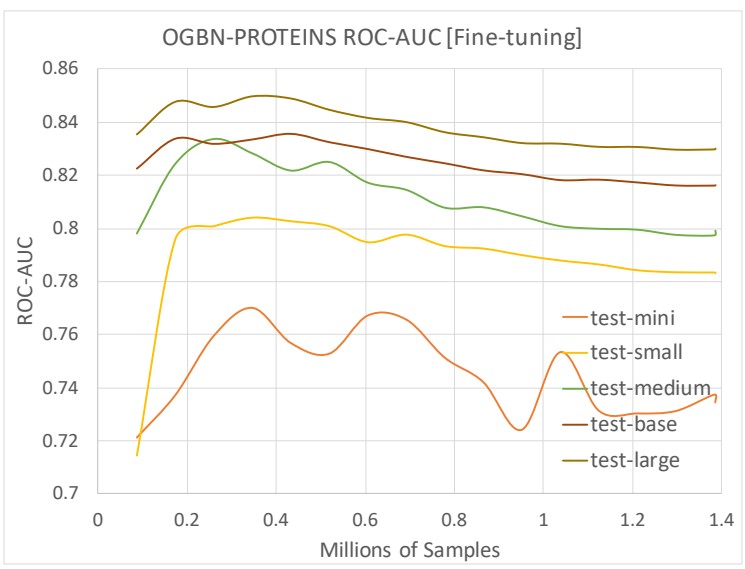

Figure 7: ROC-AUC metric on test data in the fine-tuning stage. $x$-axis is the number of samples trained during fine-tuning.

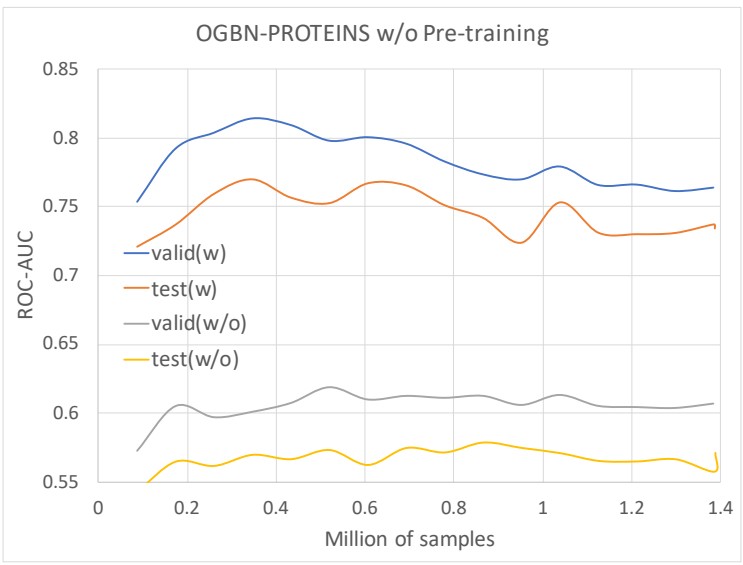

Figure 8: The ablation study of pre-training for ogbn-proteins of GraphGPT-mini. Evaluation metric ROC-AUC on the valid/test data versus the number of samples fine-tuned. 'w' and 'w/o' indicate whether pre-training is employed or not. 'w' means the model is pre-trained with 51.2 B tokens first, and then fine-tuned with the link prediction task. 'w/o' indicates that the model is trained with the supervised link prediction task directly with random parameters initialization. We use GraphGPT-mini to save time and computing resources.

Table 12: Node-level binary classification task results of ogbn-proteins dataset. The batch-size is 128, and the learning rate is $3 \times 10^{-5}$.

| Models | ROC-AUC (%) | | Params |
| --- | --- | --- | --- |
| | Test | Valid | |
| GraphGPT-mini (w/o pre-training) | 57.52±0.36 | 61.19±0.08 | 10.76M |
| GraphGPT-mini | 75.61±1.37 | 80.47±0.94 | 10.76M |
| GraphGPT-small | 80.10±0.35 | 83.36±0.40 | 29.90M |
| GraphGPT-medium | 82.71±0.52 | 86.18±0.28 | 46.68M |
| GraphGPT-base | 83.37±0.15 | 87.68±0.25 | 132.94M |
| GraphGPT-large | 84.80±0.18 | 89.35±0.24 | 428.94M |

### F.3 NODE IDENTITY ENCODING

The comparison between models trained with and without node identity encoding is shown in Fig. 9. The node identity encoding improves results.

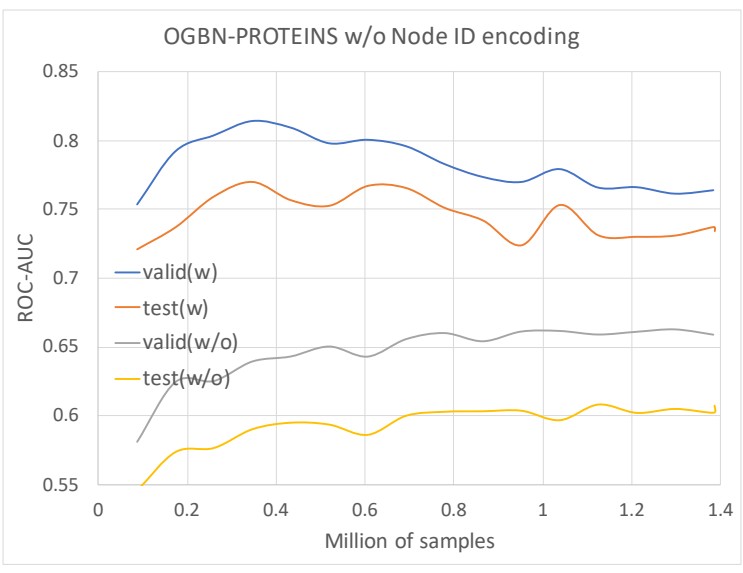

Figure 9: The ablation study of node identity encoding for ogbn-proteins of GraphGPT-mini. Evaluation metric ROC-AUC on the valid/test data versus the number of samples fine-tuned. 'w' and 'w/o' indicate whether the node is encoded with two tokens uniquely or not. Both 'w' and 'w/o' are pre-trained with 25.6 B tokens first, and then fine-tuned with the link prediction task. After fine-tuning with about 0.2 to 0.4 M samples, the results become stable for the model with node identity encoding.

## G DISCUSSION ON LIMITATIONS

We discuss some limitations of GraphGPT for a comprehensive understanding the novel model.

***Transferability.*** Due to the lack of shared semantics among different graph datasets, GraphGPT has to be pre-trained on target dataset itself and then fine-tuned. This may restrict its applicability in various graph tasks. However, there are two exceptions.

a). Structure-understanding GraphGPT. Graph structures are shared across all the graph datasets, for example, node degrees, motifs, edge directions and etc. We can pre-train a structure-understanding GraphGPT using structure information only, *i.e.*, removing all the semantics (node/edge attributes) in the graphs. This GraphGPT would be able to do any graph structure understanding related tasks,

if trained with enough data and large model size. Besides, it could be further pre-trained later on specific datasets with semantics information.

b). Domain specific GraphGPT. For example, molecular datasets share the same semantics information, *i.e.*, nodes (atoms) and edges (bonds). We can collect all available molecular datasets and pre-train a molecule-understanding GraphGPT. It can then be fine-tuned on various molecule datasets, and would have a wide application in drug and material research.

***Dataset size.*** When the graph datasets for pre-train and fine-tune are small or medium, GraphGPT might not perform well, for example, ogbn-proteins and ogbg-molpcba datasets. It could be overcome by collecting more data of the same semantics for pre-training/fine-tuning. In addition, utilizing the above mentioned 'structure-understanding GraphGPT' and further pre-training on the dataset with semantics could also be helpful.

***Compute budget.*** The pre-training on one big graph (ogbn-proteins and ogbl-ppa) and many small graphs (PCQM4M-v2) with large model sizes (50M+ parameters) are very computationally expensive.

For example, pre-training a GraphGPT-base (100M+) for PCQM4M-v2 with 25.6B tokens will take about 240 V100 GPU hours (4 V100 trained with 60 hours). Fine-tuning cost about 20 V100-hours per epoch.

For small/medium dataset, the trade-off between compute budget and performance implies that GraphGPT might not be the first choice.

However, to pursue superior performance given large amount of data, large scale GraphGPT is a very competitive candidate. Meanwhile, int8/int4-quantization techniques has been developed to allow much faster inference, even training (Dettmers et al., 2022; Frantar et al., 2022). Parallel training frameworks like DeepSpeed and Megatron enable fast mixed-precision and bf-16 precision training utilizing thousands of GPUs (Rasley et al., 2020; Shoeybi et al., 2019). Transformer-specific optimization techniques like FlashAttention-2 allows us to speed-up the training of transformer models (Dao, 2023). Last but not least, the development of GPU and tensor accelerator chips is very fast. As they are becoming much faster and cheaper, compute burden will not be a big problem.

***Context window.*** The context window of the transformer affects the computational complexity very much. This limits the efficiency of training large (sub)graphs. Besides relying on the development of efficient training of transformers (Dao, 2023), finding short sequence representation of graphs, and exploring better ways to extract sequences from big graphs are interesting to study further.

These limitations are also opportunities that worth further explorations.

