# OpenReview forum: "GraphGPT: Graph Learning with Generative Pre-trained Transformers"
_ICLR.cc/2024/Conference — Submitted to ICLR 2024_

### Official Review · Reviewer_kAz1 · 2023-10-30

**Soundness:** 2 fair
**Presentation:** 2 fair
**Contribution:** 3 good
**Rating:** 5
**Confidence:** 4

**Summary:**

The paper introduces GraphGPT, a new model for graph learning using Generative Pre-training Transformers. By converting graphs or their subgraphs to token sequences via an Eulerian path and pre-training with next-token-prediction, GraphGPT outperforms or matches state-of-the-art methods on various datasets, for tasks at graph, edge, and node levels, including PCQM4Mv2, ogbl-ppa, and ogbn-proteins from OGB. Its generative pre-training allows training with 400M+ parameters with increasing performance, surpassing the capability of GNNs and previous graph transformers.

**Strengths:**

- The paper presents a compelling and effective approach to converting graph learning problems into NLP problems, which is an innovative bridge between the domains of NLP and graph learning.
- The method showcased state-of-the-art performance, specifically on the ogbl-ppa link prediction dataset.

**Weaknesses:**

1. **Discussion on limitations**: The paper does not delve into potential limitations of the proposed method.
2. **Reference gap**: Despite the extensive literature on graph transformers, like the works of Rampášek et al. 2022 and Chen et al. 2022, the paper cites only a few, leading to a lack of comprehensive context.
3. **Computational concerns**: The proposed method, while achieving good performance on some datasets, appears to demand significant computational resources. There is a clear lack of discussion on the resources' requirements and the computation time compared to other graph models.
4. **Transferability issues**: GraphGPT has to be re-trained for each specific dataset, raising concerns about its adaptability across diverse graph modalities or sizes. This constraint, paired with the computational burden of pre-training a large GraphGPT, casts doubts on its practical applicability.
5. **Performance claims**: The paper might be overstating its model's performance. While it excels in the ogbl-ppa link prediction dataset, its performance appears mediocre in both graph and node-level prediction tasks when compared against similar scale GNN/GT baseline models.

_References:_

- Rampášek, Ladislav, et al. "Recipe for a general, powerful, scalable graph transformer." NeurIPs 2022.
- Chen, Dexiong, Leslie O’Bray, and Karsten Borgwardt. "Structure-aware transformer for graph representation learning." ICML 2022.

**Questions:**

**Q1**: What factors contribute to the model's superior performance in the ogbl-ppa link prediction dataset, yet not replicating similar success in node and graph-level prediction tasks? Some intuitive explanation would be helpful.

I will be happy to increase my rating if the authors can address all my concerns and questions

---

> ### Author Response · Authors · 2023-11-22
> **Discussion on limitations and performance**
>
> W1: Discussion on limitations: The paper does not delve into potential limitations of the proposed method.
>
> A: Thank you for pointing out. We will add the following discussion on limitations in the paper:
> 1.	Transferability: Due to the lack of shared semantics among different graph datasets, GraphGPT has to be pre-trained on target dataset itself and then fine-tuned. However, there are two exceptions.
>
> a). Structure-understanding GraphGPT. Graph structures are shared across all the graph datasets, for example, node degrees, motifs, edge directions and etc. We can pre-train a structure-understanding GraphGPT using structure information only, i.e., removing all the semantics (node/edge attributes) in the graphs. This GraphGPT would be able to do any graph structure understanding related tasks, if trained with enough data and large model. Besides, it could be continued pre-trained later on specific dataset with semantics information.
>
> b). Domain specific GraphGPT. For example, small organic molecules datasets share the same semantics information, i.e., nodes (atoms) and edges (bonds). We can collect all available molecules datasets and pre-train a molecule-understanding GraphGPT. It would have a wide application in drug and material research.
> 2.	Dataset size: When the graph dataset for pre-train and fine-tune is small or medium, GraphGPT might not perform well. For example, ogbn-proteins and ogbg-molpcba datasets. It could be overcome by collecting more data of the same semantics for pre-training/fine-tuning. Also, continue pre-training on above mentioned structure-understanding GraphGPT could also be a direction to explore.
> 3.	Compute budget: The pre-training on one big graph (ogbn-proteins and ogbl-ppa) and many small graphs (PCQM4M-v2) with large models (50M+ params) are very compute expensive.
>
> For example, pre-train a GraphGPT-base (100M+) for PCQM4M-v2 with 25.6 B tokens will take about 240 V100 GPU hours (4 V100 trained with 60 hours). Fine-tune cost 20 V100-hours per epoch.
>
> For small/medium dataset, the trade-off between compute budget and performance may imply that GraphGPT is not the first choice.
>
> However, to pursue superior performance given large amount of data, GraphGPT with large number of parameters is a very competitive candidate.
>
> Meanwhile, int8/int4-quantization techniques* has been developed to allow much faster inference, even training. Parallel training frameworks like DeepSpeed and Megatron enable fast mixed-precision and bf-16 precision training utilizing thousands of GPUs. Transformer-specific optimization techniques like FlashAttention-2** allows us to speed-up the training of transformer models. Last but not least, the development of tensor accelerator chips is very fast. As they are becoming much faster and cheaper, compute burden won’t be a big problem.
>
> * 8-bit Optimizers via Block-wise Quantization
> * GPTQ: Accurate Post-Training Quantization for Generative Pre-trained Transformers
> ** FlashAttention-2: Faster Attention with Better Parallelism and Work Partitioning
>
>
> 4.	Context window: The context window of transformer affects the compute burden very much, so it limits the efficiency of training large (sub)graphs. Besides relying on the development of efficient training of transformers (e.g., FlashAttention-2), finding short sequence representation of graphs, and exploring better ways to extract sequences from big graphs are interesting to study further.
>
> These limitations worth further explorations, and might serve as potential research directions.
>
> W2: Reference gap
>
> A: Thank you for pointing out. We will add them in the reference.
>
> W3: Computational concerns
>
> A: Please refer to the above discussion on limitations about compute budget.
>
> W4: Transferability issues
>
> A: Please refer to the above discussion on limitations about transferability.
>
> W5: Performance claims
>
> A: Our performance on the large scale graph-level task PCQM4M-v2 is on par with the SOTA given the comparable number of parameters.

---

> ### Author Response · Authors · 2023-11-22
> **Analysis of performance gap among different graph tasks**
>
> Q1: Performance gap among edge-/node-/graph-level tasks
>
> A: The performance of GraphGPT on edge-level dataset ogbl-ppa and graph-level dataset PCQM4M-v2 are good, while on node-level ogbn-proteins and ogbg-molpcba below SOTA.
>
> 1. Dataset size: ogbl-ppa and PCQM4M-v2 are large for pre-train and fine-tune. For example, ogbl-ppa’s pre-train and fine-tune data is actually very huge. The positive samples are 30M+ edges, and the negative samples are randomly sampled node pairs (i.e., negative edges) of scale 576,2892 ~ 332 B. This will allow the large model to be well trained without much over-fitting. At the same time, PCQM4M-v2 is pre-trained with 3.7M+ molecules, and fine-tuned with 3.38M.
>
> In contrast, ogbn-proteins’ fine-tune dataset is only about 87K, and ogbg-molpcba’s pre-train and fine-tune dataset is only about 0.4M.
>
> 2. Consistency: For ogbl-ppa, the pre-training and fine-tuning tasks are close. In pre-training, NTP predict next attributes and next nodes. Predicting next nodes can be regarded as link prediction, which is aligned with the fine-tune task.
>
> In contrast, for ogbn-proteins, the NTP that predicts the node’ attributes is similar to the fine-tune task of predicting node properties. However, the node attribute is the 8 species, which is simple so that model cannot learn enough information relevant to fine-tune task during pre-training.
>
> 3. Please refer to the reply to Q1 and Q2 of reviewer `VKoy` for additional analysis.

---

> > ### Comment · Reviewer_kAz1 · 2023-12-03
> > **Comment after rebuttal**
> >
> > I appreciate the authors' clarifications provided for my questions. Nonetheless, I must express that I remain unsatisfied with the response. My concerns about the computational aspects and the transferability issues persist as significant drawbacks of the suggested approach, as also acknowledged by the authors. More importantly, in terms of performance on the graph-level task PCQM4M-v2, it is evident that the proposed method falls short when compared to the state-of-the-art graph transformers. For instance, GPS introduced by Rampášek 2022, with only 13M parameters, achieved a MAE of 0.0852, significantly surpassing GraphGPT's 0.0875. Consequently, I keep my initial rating.

---

### Official Review · Reviewer_VKoy · 2023-10-31

**Soundness:** 3 good
**Presentation:** 3 good
**Contribution:** 3 good
**Rating:** 5
**Confidence:** 3

**Summary:**

This paper presents GraphGPT, an adaptation of the Transformer network tailored for graph data. GraphGPT is adept at handling a wide range of graph datasets and various graph-related tasks, including node, edge, and graph prediction. A notable innovation in this work is the use of (semi-)Eulerian paths to transform graphs into sequences of tokens. This transformation is designed to be lossless and reversible, ensuring the integrity of the original graph data. GraphGPT is pre-trained on the NTP task and is fully compatible with the Transformer's decoding architecture. This compatibility allows GraphGPT to fully leverage the benefits of generative pre-training. Empirical results showcased in the paper highlight GraphGPT's performance, achieving near or on par with state-of-the-art results in tasks at graph level, edge-level, and node level.

**Strengths:**

The paper is well-written and logically structured. The application of (semi-)Eulerian paths is a  novel idea, demonstrating considerable versatility in converting graph structures into decoding sequences. The diverse empirical results underscore the adaptability of GraphGPT, showcasing its ability to yield competitive performance across a range of traditional graph tasks.

**Weaknesses:**

The empirical results highlight a critical aspect of GraphGPT, especially in terms of its performance relative to the number of parameters. While the approach is versatile and straightforward, making it adaptable for various tasks, I find its substantial size problematic. The graph-level and node-level task performances, in my opinion, don't seem to justify the significantly larger scale of GraphGPT compared to its competitors.

I recognize the primary advantage of GraphGPT as its simplicity in application across different graph tasks. However, I question the rationale for preferring GraphGPT in scenarios other than edge-level tasks. While alternative methods may be more intricate and rely on "tricks," they often prove to be more efficient and effective.

Moreover, I am skeptical of the assertion that GraphGPT is immune to over-smoothing and over-squashing, based solely on the moderate performance improvements observed with increased parameter size. This claim demands further scrutiny, particularly when considering the relatively poor performance in node classification tasks.

I believe that a more comprehensive investigation is required, especially focusing on how node parameters are represented. It's essential to determine whether the modest improvements in performance are indeed due to the avoidance of over-smoothing and over-squashing or whether other factors are at play.

**Questions:**

see weaknesses

---

> ### Author Response · Authors · 2023-11-22
> **performance concerns and over-smoothing**
>
> Q1: performance on graph/node-level tasks and scalability
>
> A: The performance of GraphGPT on large-scale graph-level dataset PCQM4M-v2 (3.7m) is on par with the SOTA with the similar number of parameters.
>
> The performance on the medium-scale graph-level dataset ogbg-molpcba (0.4m) is below SOTA, the reason could be two:
>
> 1.	The dataset is still `small`. Pre-trained on itself only cannot let the large model gains much knowledge of molecules before fine-tuning. In fine-tune stage, small training data + non-well-pre-trained large model, will easily overfit, and lead to below SOTA results. Practically, it has been demonstrated many times that handcrafted features + simple model usually outperforms large models given limited data.
>
> 2.	Optimization. Unlike NLP, transformer on molecule data is not well studied yet. So the best optimization strategies remain to be explored. For example, the optimizer and its hyper-parameters, the regularization methods and etc. Recently we found that gradient-clipping that commonly used in NLP hurts the performance of GraphGPT very much.
>
> Based on the practice of Graphormer, to perform well in ogbg-molpcba, it is pre-trained on the PCQM4M’s 3.7m+ supervised samples. So, together with our preliminary experiments, we hypothesize that GraphGPT on ogbg-molpcba could be further improved when pre-trained with NTP on more molecules data, and better optimization hyper-parameters.
>
>
> Q2: over-smoothing and over-squashing
>
> A: Over-smoothing points out that GNN’s nodes’ output embeddings tend to be the same when the number of layers increases. It restricts GNN to be usually less than 10 layers and much less than 1m parameters. When go deeper, the performance of GNNs drops significantly.
>
>   In contrast, our GraphGPT-large has 24 layers and 400m parameters, much deeper and larger than GNNs. More importantly, our models’ performance keeps increase, which is clearly not the sign of over-smoothing, although its performance is still below SOTA.
>
>    In addition, during pre-training with NTP, each node’s output embedding will be used to calculate the cross-entropy loss with the embeddings of next node or attributes. If these embeddings are close, the training loss will not decrease and the model cannot be trained. From pre-train loss curve, we can see that large model yields lower training loss, and it is still decreasing after trained with billions of tokens.
>
> Over-squashing stats that long-range important information will diminish much due to the aggregating mechanism of GNNs. GraphGPT’s global self-attention can easily overcome this limitation, which has been well studied in NLP and CV.
>
>    The main reasons of node-level task ogbn-proteins below SOTA could as follows:
>
> 1.	Training data too small in fine-tune stage. This dataset has about 132k nodes, and 86.6k are used to training. It’s very small compare to the model size, so it can be easily overfitting.
>
> 2.	Information (feature) per-sample is too little. In GraphGPT, each node sample can only extract information from 9 neighbors when using context window 256 (Tab. 9 in App. A1). In contrast, SOTA GNNs partition the graph into 10, each partition contains 8.7k nodes, and the node can gather information from all the nodes in the same partition during training. The limited context window restrict GraphGPT attending many neighbor nodes, but it can be overcome as more efficient training techniques of transformer with larger context window are developed, for example, FlashAttention-2*.
>
> * FlashAttention-2: Faster Attention with Better Parallelism and Work Partitioning

---

> > ### Comment · Reviewer_VKoy · 2023-11-22
> > **Responding the author's reply**
> >
> > Thank you for providing explanations to my questions. However, I find the explanations unsatisfactory. Attributing the poor performance to the size of the dataset and the optimization scheme as excuses only further highlights the paper's lack of readiness. Moreover, using the size of GraphGPT to justify the absence of over-smoothing or over-squashing without the support of additional studies is highly naive. For these reasons, I will downgrade my initial assessment to a 3.

---

### Official Review · Reviewer_2YEp · 2023-11-02

**Soundness:** 2 fair
**Presentation:** 3 good
**Contribution:** 2 fair
**Rating:** 3
**Confidence:** 3

**Summary:**

This paper proposes GraphGPT, a novel model for graph representation learning. The key ideas are: 1) Transform graphs into sequences of tokens via Eulerian paths to preserve structure information. 2) Pretrain the transformer decoder with next token prediction on the graph sequences. 3) Fine-tune on downstream supervised graph tasks by formatting them to be compatible with the decoder. Experiments on graph, edge and node classification tasks demonstrate strong performance and consistently improving results when scaling up GraphGPT.

**Strengths:**

- The graph-to-sequence transformation using Eulerian paths is an elegant way to serialize graphs for the transformer. This avoids complex feature engineering.
- Leveraging generative pretraining enables scaling up to hundreds of millions of parameters, overcoming limitations of GNNs.
- GraphGPT consistently improves with more data and parameters, demonstrating generalization ability.

**Weaknesses:**

- Performance on some node and edge tasks is not state-of-the-art.
- Limited analysis of what properties are learned during pretraining and their utility.
- Does not experiment with very large models in the billions of parameters range.

**Questions:**

- For large graphs, how are the subgraph sampling parameters (depth, context size) chosen? Is there a principled way to set these?
- What is the computational complexity of the graph-to-sequence transformation? How does it scale?
- Is the decoder-only transformer architecture sufficient for graph tasks or would an encoder-decoder be beneficial?

---

> ### Author Response · Authors · 2023-11-22
> **Performance concerns and billion-scale models**
>
> W1: Performance on some node and edge tasks is not state-of-the-art.
>
> A: Although the performance on some tasks are not SOTA yet, we still present it in the paper, for 2 reasons.
>
> 1. GraphGPT’s potential. Although the results are not SOTA, they are already quite close. As proved in NLP and CV, pre-train with more data and large model can consistently improve the performance in downstream tasks. So we expect GraphGPT to archive SOTA when scaling up the data-size and model-size, given that it already attains close to SOTA results currently. In this work, we mainly show GraphGPT’s promising potential in various graph tasks.
>
> 2. GraphGPT’s limitation. As we pointed out in the appendix, GraphGPT does not perform well in ogbl-ddi dataset. We present some intuitive analysis on its limitation. For example, when the graph is small and very dense, GraphGPT may not be a suitable choice. If the dataset is small, GraphGPT cannot learn much useful information from pre-training. And for very dense graph, the structure may not be able to provide enough information during pre-training.
>
> For some additional analysis on why GraphGPT’s results are not SOTA in these datasets, please refer to the reply to Q1 and Q2 of reviewer `VKoy`.
>
>
> W2: Limited analysis of what properties are learned during pretraining and their utility.
>
> A: Thank you for pointing out. In the paper, we pre-train the model with NTP task to generate the graphs, and together with the structure and semantics information. So the model is expected to learn the overall distribution of graphs, and their structures and semantics.
>
> The fine details of pre-training, such as the different roles played by the structure and semantics information are not studied in this work, and it is indeed an interesting topic worth further studies. For example, we can pre-train models with graph’s structure-information-only vs structure-semantics-info, and see how the fine-tuning results would be different. However, the overall usefulness of pre-training has been demonstrated in the ablation study, which is the main point that we want to emphasize.
>
>
> W3: Does not experiment with very large models in the billions of parameters range.
>
> A: There are several limitations that prevents us from scaling up to billion-scale in this work.
>
> 1. Resource limitation: billion-scale model consumes huge amount of computing resources and very time-consuming. Before we have a better understanding of GraphGPT in relatively small scale and try to extrapolate to larger scale, we cannot afford the training budget. (We definitely will scale up to billion-scale in the future works.)
>
> 2. Dataset limitation: current graph datasets cannot provide enough amount of diverse data to be feed to billion scale models.
>
> 3. Optimization limitation: In NLP, before the emergence of billion-scale models, lots of work has been done to find suitable optimization settings, such as batch-size, lr, lr scheduler, dropout rate, optimizer and optimizer hps, tokenization methods, and etc.
>
> The serialized graph data is analogous to text, but they are essentially different. For example, the molecular graphs have a vocabulary size about hundreds, while in NLP the vocab is usually above 30k (Llama’s vocab is 32k). The tokens in GraphGPT are node-id, node/edge-attributes and etc, and they are quite different in natural. In NLP the tokens are word pieces. Therefore, the optimal optimization settings could be very different from NLP.
>
> As the 1st worker to serialize graph using Eulerian path and pre-train with NTP, the optimal optimization strategy remains to be further studied before diving into billion-scale models.

---

> ### Author Response · Authors · 2023-11-22
> **Reply to questions of subgraph sampling, computational complexity and architecture**
>
> Q1: For large graphs, how are the subgraph sampling parameters (depth, context size) chosen? Is there a principled way to set these?
>
> A: The parameters are mainly determined by the context window of transformer. For example, if we have large compute budget, we could set it to be 1024. Then we run 10k subgraph samplings with DFS to find the depth such that 90% of the resulting sequences are within the context window. We also tried BFS to find the appropriate number of neighbors (with depth fix to 1).
>
>  Empirically, we experiment with DFS (with num_neighbors=1) and BFS (with depth=1) on ogbn-proteins, and doesn’t obtain much different results. In principle, if given enough compute budget, larger subgraphs might yield better results. We preliminarily experiment with context window from 256 to 1024 in this dataset, but does not gain significant performance gains. It could be due to the ease of overfitting and small dataset size.
>
> For edge-level dataset ogbl-ppa, previous works show that the neighbors play important role in the link prediction task, so we choose BFS to sample the immediate neighbors.
>
>
> Q2: What is the computational complexity of the graph-to-sequence transformation? How does it scale?
>
> A: The most time consuming part of the transformation is the Eulerization and Finding Eulerian paths.
> 1.	Eulerization’s time complexity is O(n3)*, where n is the number of nodes of odd degrees.
> 2.	Finding Eulerian paths’ complexity is O(|E|)**, where |E| is the number of edges.
>
> We use the implementation in `networkx` package. During the training and inference, the Eulerization and finding Eulerian paths are calculated on-the-fly. And it is not the bottle-neck.
>
> * https://cs.stackexchange.com/a/9129
> * https://en.wikipedia.org/wiki/Eulerian_path#Hierholzer.27s_algorithm
>
>
> Q3: Is the decoder-only transformer architecture sufficient for graph tasks or would an encoder-decoder be beneficial?
>
> A: In NLP, decoder-only (such as GPT-series) and encoder-decoder (such as T5) do not show significant difference as measured by various downstream tasks. So we hypothesize that they might behavior similarly in graph tasks, and due to the limitation of various resources, we finally do not study both of them in this work.
>
> However, it worth exploring their different behaviors in graph tasks. In addition, encoder-only architecture (such as Bert) usually out-performs decoder-only architecture when they are restricted to the small/medium-size dataset and same model size. So, comparing the 3 architectures under different datasets and model sizes settings would be interesting to explore, but it is out of the scope of this work.

---

### Official Review · Reviewer_v4XD · 2023-11-05

**Soundness:** 3 good
**Presentation:** 3 good
**Contribution:** 2 fair
**Rating:** 5
**Confidence:** 3

**Summary:**

The paper proposes  a novel model for Graph learning by self-supervised Generative Pre-training Transformers. The proposed method includes: 1) transforming the (sub)graphs into a reversible sequence of tokens via the Eulerian path, 2) pre-training a transformer decoder using the NTP task, and 3) fine-tuning the transformer. The paper investigates various graph related tasks: graph-/edge-/node-level tasks

**Strengths:**

The paper proposed an interesting angle for graph pretraining, which takes advantage of the great performance of the breakthrough of LLM (transformer) into graph learning. The idea of Eulerian path is also interesting. The paper has investigated various graph tasks, including graph-/edge-/node-level tasks; and also consider small/large graph

**Weaknesses:**

-- Due to high variance of graph benchmakrs, to prove the effectiveness of the proposed methods (graph pretraining & finetuning), I would experct the authors provide more benchmarks results.

-- There are various graph pretraining methods proposed (i.e. either combined with transformer or not, use contrastive learning or not, etc). Can the author compared with various other pretraining methods?

-- The ablation study of pretraining is helpful. i.e. table 5. Just curious, are their other ablation study on finetuning?

**Questions:**

See above.

---

> ### Author Response · Authors · 2023-11-22
> **More benchmarks and pre-training methods comparison**
>
> Q1: Due to high variance of graph benchmakrs, to prove the effectiveness of the proposed methods (graph pretraining & finetuning), I would expect the authors provide more benchmarks results.
>
> A: There are several reasons that we didn’t experiment on more benchmarks.
>
> 1.	Dataset size. Most graph datasets are very small, so large models are usually not applicable. As we see in NLP and CV, large models + pre-train & fine-tune paradigm excels mostly when the dataset is very huge.
>
> 2.	Unrelated semantics. In NLP or CV, different datasets share the same underlying semantics. So, Bert or GPT can just pre-train once and then fine-tune on many down-stream tasks. In graph datasets, the only common feature is the `graph structure’, but their semantics are very different. Therefore, we cannot pre-train one GraphGPT on the collection of datasets and then fine-tune. GraphGPT has to be pre-trained and then fine-tuned on every benchmark individually. This is very time-/resource- consuming. (Molecular datasets are the exception, as they share the same semantics, i.e., the same atoms and bonds. But training a domain-specific GraphGPT is not the theme of this work.)
>
> As the 1st work of this new direction, besides some SOTA results, we want to show GraphGPT’s potential on solving various large-scale graph problems. Therefore, we have already chosen the popular graph-/edge-/node-tasks datasets that are very large in graph domain.
> However, these datasets are still small compare to datasets in NLP and CV.
>
>
> Q2: There are various graph pretraining methods proposed (i.e. either combined with transformer or not, use contrastive learning or not, etc). Can the author compared with various other pretraining methods?
>
> A: In the dataset we studied, the top performing models do not involve pre-training like contrastive learning. In contrast, some methods employs transfer learning, i.e., pre-trained with supervised signal from external data. For example, in ogbg-molpcba leaderboard, the top two methods HIG and Graphormer are trained using PCQM4M’s datasets and its supervised tasks. Some methods in PCQM4M-v2 leaderboard are pre-trained with external 3D conformation data calculated by `rdkit`. Therefore, there might not be suitable pre-training methods to be compared for these datasets we stuidied.
>
>
> Q3: The ablation study of pretraining is helpful. i.e. table 5. Just curious, are their other ablation study on finetuning?
>
> A: In our setting, zero-shot is not applicable, so fine-tuning is necessary. In fine-tuning, we tried different hyper-parameters, such as drop-out rate, weight-decay rate, learning rate scheduler and etc. These are common techniques, so we didn’t study them in ablation in the paper.

---

### Meta-Review · Area_Chair_u3HM · 2023-12-06

**Metareview:**

This paper introduces a graph representation learning model which transforms graphs into token sequences. To ensure structural information is maintained, Eurlerian paths are used during this transformation. The sequences are then used to pretrain a transformer decoder and then to fine-tune on a supervised task.

Key strengths:
- The paper is well-written and the key concepts are explained well
- Using Eulerian paths is both interesting and novel in this context
- The high-level idea of using generative pretraining in the graph tasks is compelling

Key weaknesses:
- Experimental setup: Almost all reviewers are not convinced by the experimental setup, mentioning insufficient benchmarks and comparisons.
- Experimental results: The results are not very compelling. This is not to say that every paper needs to demonstrate SOTA results, but in this case there are incomplete insights about the scenarios where this method could improve performance and what are the factors hindering its performance in certain datasets.
- Justification of the method: overall the idea makes sense intuitively but the claims are not accompanied by convincing results or insights, raising the question: why should one use this method instead of a GNN? For example, two reviewers mention concerns about versatility, while others are not convinced about claims regarding improvements for oversmoothing / oversquashing.

The reviewers feel that the rebuttal has not addressed their concerns adequately and I tend to agree.

**Justification For Why Not Higher Score:**

The rebuttal does not address reviewers' concerns satisfactorily. For most concerns the authors offer intuitive explanations which are not convincing enough, and for other concerns the rebuttal points to future work.

**Justification For Why Not Lower Score:**

N/A

---

### Decision · Program_Chairs · 2024-01-16

Reject